# Effects of Dietary N-Carbamylglutamate on Growth Performance, Apparent Digestibility, Nitrogen Metabolism and Plasma Metabolites of Fattening Holstein Bulls

**DOI:** 10.3390/ani11010126

**Published:** 2021-01-08

**Authors:** Jinshan Yang, Jian Zheng, Xinpeng Fang, Xin Jiang, Yukun Sun, Yonggen Zhang

**Affiliations:** College of Animal Science and Technology, Northeast Agricultural University, Harbin 150030, China; yjs853396960@sina.com (J.Y.); wasj315@126.com (J.Z.); dnfangxinpeng@163.com (X.F.); prozyg@sina.com (X.J.); sun_yukun@126.com (Y.S.)

**Keywords:** *N*-carbamylglutamate, bulls, growth, nitrogen utilization, arginine

## Abstract

**Simple Summary:**

It is critical to find an effective and safe feed additive to improve the utilization of nitrogen by ruminants. *N*-carbamylglutamate (NCG), a structural analogue of *N*-acetylglutamate, may have the potential to improve the utilization of nitrogen by bulls. Therefore, the effects of adding different levels of NCG on the growth performance, nutrient digestibility, nitrogen metabolism and plasma metabolites of fattening Holstein bulls were investigated in this study. The addition of NCG increased concentrations of plasma Arg, Ile, Val, total essential amino acid and total nonessential amino acid, which in turn resulted in higher *N* utilization and CP digestibility for growth and, therefore, higher average daily gain in NCG-fed bulls.

**Abstract:**

*N*-carbamylglutamate (NCG), a structural analog of *N*-acetylglutamate, improves nitrogen utilization in dairy cows. However, the effects of NCG on bulls are unknown. The purpose of the current research was to investigate the effects of adding different amounts of NCG on growth performance, nutrient digestibility, nitrogen metabolism and plasma metabolites of fattening Holstein bulls. Twenty-four Holstein bulls with similar body weights (BW, 408 ± 21.9 kg) and ages (450 ± 6.1 d; all mean ± SD) were selected for the feeding trial. After 2 weeks of adaptation, bulls were blocked by BW and age and subsequently randomly assigned to 1 of 4 groups: (1) CON group (control diet), (2) L group (supplementation with 20 mg/kg BW NCG), (3) M group (supplementation with 40 mg/kg BW NCG), or (4) H group (supplementation with 80 mg/kg BW NCG). The addition of NCG linearly and quadratically increased the average daily gain (CON vs. L vs. M vs. H = 1.03 vs. 1.19 vs. 1.40 vs. 1.26 kg/d) (*p* < 0.05), feed conversion ratio (CON vs. L vs. M vs. H = 11.92 vs. 9.22 vs. 7.76 vs. 8.62) (*p* < 0.05), crude protein digestibility (CON vs. L vs. M vs. H = 64.3 vs. 63.8 vs. 67.7 vs. 65.8%) (0.05 < *p* < 0.10), N retention (*p* < 0.05) and *N* utilization (*p* < 0.05) of bulls, whereas the contents of fecal *N* (0.05 < *p* < 0.10) and urinary *N* (0.05 < *p* < 0.10) in NCG-fed bulls linearly decreased compared with those in CON bulls. Bulls fed NCG showed a quadratic increased plasma nitric oxide (*p* < 0.05) concentration. Furthermore, Arg (*p* < 0.05), Ile (*p* < 0.05), Val (*p* < 0.05), Ala (*p* < 0.05), Glu (*p* < 0.05), Ser (*p* < 0.05), total essential amino acid (*p* < 0.05) and total nonessential amino acid (*p* < 0.05) concentrations linearly and quadratically increased with increasing doses of NCG. In contrast, plasma urea (*p* < 0.05) and ammonia (*p* < 0.05) concentration linearly and quadratically decreased with increasing doses of NCG. Overall, the addition of NCG increased plasma Arg, Ile, Val, TEAA and TNEAA concentration, which in turn resulted in a higher N utilization and, therefore, higher average daily gain in NCG-fed bulls, providing baseline data for the widespread application of NCG in beef cattle production.

## 1. Introduction

Nitrogen plays a critical role in the growth and development of animals [1]. Roughage and non-protein nitrogen sources, which are not digested by humans, can be converted into meat and milk by ruminants [2]. However, nitrogen in feed cannot be fully utilized by ruminants, with approximately 70% of that ingested being excreted in the feces and urine [3,4]. This not only causes environmental pollution but also limits the competitiveness of livestock farms. Therefore, it is critical to find an effective and safe feed additive to improve nitrogen utilization by ruminants.

Arginine (Arg), a nutritional amino acid, is involved in nitrogen metabolism by promoting the synthesis of urea and nitric oxide (NO) [5]. However, it is difficult for it to reach the small intestine and carry out biological functions due to its rapid degradation in rumen. [6]. A previous study demonstrated that infusions of Arg through the jugular vein could improve the lactation performance and nitrogen utilization of dairy cows [7]. Interestingly, *N*-carbamylglutamate (NCG), a structural analog of *N*-acetylglutamate (NAG), can activate the rate-limiting carbamoyl-phosphate synthase in the urea cycle and arginine synthesis pathway and promote the synthesis of endogenous arginine [8,9]. A previous study found that NCG has a much lower degradation rate than arginine in the rumen [6], indicating that NCG could better reach the small intestine and carry out biological functions. Zhang et al. (2018) reported that dietary NCG improved growth performance, intestinal integrity, immune function and antioxidant capacity of suckling lambs [10]. Gu et al. (2018) showed that milk fat and protein contents of dairy cows were improved with NCG supplementation [11]. Chacher et al. (2014) demonstrated that NCG addition optimized the amino acids profile and improved the nitrogen utilization by reducing the urea nitrogen content in milk and plasma in high-producing dairy cows [12]. These studies indicate that NCG has the potential to promote production performance and health of ruminants by regulating nitrogen metabolism.

However, to date, the effects of NCG on nitrogen metabolism in beef cattle have not been evaluated. Our hypothesis for this study was that NCG may be used in the diets of beef cattle to improve their utilization of the nitrogen. Therefore, the objective of the current research was to investigate the effects of adding different levels of NCG on the growth performance, nutrient digestibility, nitrogen metabolism, plasma metabolites and blood amino acids of fattening Holstein bulls.

## 2. Materials and Methods

### 2.1. Animals, Experimental Design and Diets

This experiment was conducted at the Acheng Test Base of Northeast Agricultural University Harbin, China, from November 2018 to January 2019. The study was performed in strict accordance with the recommendations of the National Research Council Guide, and all animal experimental procedures were approved by the Northeast Agricultural University Animal Science and Technology College Animal Care and Use Committee (Protocol number: NEAU-[2011]-9).

Twenty-four healthy Holstein bulls with body weights (BW, 408 ± 21.9 kg) and ages (450 ± 6.1 d, all mean ± SD) were used for the feeding trial using a completely randomized block design was used in this trial [13]. Bulls were housed in individual pens (2.5 × 3 m^2^) on floor bedding with a front metal gate allowing access to feed and water. After 2 weeks of adaptation, bulls were categorized by BW and age and subsequently randomly assigned to 1 of 4 groups: (1) CON group (Control diet), (2) L group (supplementation with 20 mg/kg BW NCG), (3) M group (supplementation with 40 mg/kg BW NCG), and (4) H group (supplementation with 80 mg/kg BW NCG). NCG was mixed with 500 g diets (ingredients and nutrient compositions are presented in Table 1) and subsequently fed to bulls at 0600 every day. Bulls were weighed every week to adjust the NCG supplementation. Diets (Table 1) were formulated using the Beef Cattle Nutrient Requirements Model (2016) to meet the animals’ nutritional requirements of bulls with 1.33 kg/d average daily gain. [14]. Animals were fed twice per day at 0600 h and 1800 h. The refusals of each day were collected every morning before feeding and weighed. Based on the feed intake of the bull the day before, the amount of feed offered was adjusted daily to allow for at least 5% refusal (on an as-fed basis). Bulls had free access to freshwater throughout the trial.

### 2.2. Data Measurement and Sample Collection

After 2 weeks of adaptation, the feeding trial was performed and lasted for 7 weeks. Data measurement and sample collection were carried out at weeks 0, 4 and 7 of the trial. Body weights of bulls were the average of 3 consecutive weights at weeks 0, 4 and 7 of the trial, and thereafter, the average daily gain (ADG) was calculated. The diet offered and refused for individual bulls was weighed every day throughout the trial to calculate dry matter intake (DMI). Samples of individual feed ingredients, orts, and diets were collected weekly during the experimental period and stored at −20 °C [15]. Thereafter, all the feed samples were dried at 60 °C for 48 h, milled to pass through a 1 mm screen, and stored in sealed plastic bags with a size of 150 mm × 220 mm at 4 °C until chemical composition determination. On the last three days of weeks 0, 4 and 7 in the trial, approximately 500 g of fecal samples were taken from the rectum at 0600 and 1800 h, and samples were pooled to obtain a composite for individual bulls in 0, 4 and 7 weeks, respectively. These samples were dried at 60 °C for 48 h, milled to pass through a 1 mm screen and stored in sealed plastic bags at 4 °C until nutrient digestibility determination.

Coccygeal vein/artery blood was collected by sodium heparin tubes 3 h before the morning feeding on 3 consecutive days at weeks 0, 4 and 7 of the trial. Subsequently, blood samples were centrifuged at 2000× g for 15 min at 4 °C to obtain plasma and stored at −20 °C until determination of plasma metabolites. On the last three days of weeks 0, 4 and 7 in the trial, urine samples (10 mL) were collected from all bulls during urination with stimulation at 0600 and 1800 h, and samples were pooled to obtain a composite for individual bulls in 0, 4 and 7 weeks, respectively. The urine was acidified immediately by mixing it with 40 mL of H_2_SO_4_ (0.036 mol/L) and stored at −20 °C for future determination of urine *N* output. Valadares et al. (1999) and Leonardi et al. (2003) have demonstrated that creatinine can be used as a marker to estimate urine volume [16,17]. In calculating urine volume, we assumed creatinine output averages 29 mg/kg of BW as estimated [18]. Creatinine was detected by a colorimetric picric acid method (Shingfield and Offer, 1999) [19]. Before starting the determination, urine samples were thawed and composited (10 mL) by volume for each bull during the trial period.

### 2.3. Analysis of Feed Nutrient Composition

Samples of individual feed ingredients, orts and diets were sent to the Animal Nutrition Laboratory of Northeast Agricultural University (Harbin, China) for nutrient analysis using wet chemistry methods. The dry matter (DM, method 934.01), ash (method 938.08), crude protein (CP, method 954.01), and ether extract (EE, method 920.39) contents of the samples were determined according to the procedures of the Association of Official Analytical Chemist (2000) [20]. The heat-stable α-amylase-treated neutral detergent fiber (NDF) and acid detergent fiber (ADF) contents were analyzed according to the procedure described by Van Soest et al. (1991) [21].

### 2.4. Analysis of Nutrient Digestibility and Nitrogen Metabolism

The total tract apparent nutrient digestibility was estimated using the concentration of indigestible NDF (iNDF) in the diet and feces as an internal marker as reported by Lee and Hristov (2013) [22]. The iNDF content in the feces, feed and orts was determined by a 288 h in situ incubation, as detailed by Tosta et al. [23]. The DM, CP, NDF and ADF contents of fecal samples were analyzed using the same procedures as described for the feeds [20,21]. Digestibility of DM was calculated as digestibility of DM (%) = [1−(% of iNDF intake/% of iNDF in feces)] × 100. The digestibility of nutrients was calculated as digestibility of nutrients (%) = {1−[(% of iNDF intake/% of iNDF in feces) × (% of nutrient in feces/% of nutrient intake)]} × 100. Fecal *N* (g/d) = [CP intake (g/d)-CP intake (g/d) × % of CP Digestibility]/(% of CP in feces)/6.25 [18]. Urine *N* (g/d) = urine volume (L/d) × [CP in urine (g/L)]/6.25 [18]. Nitrogen retention was calculated as N retention (g/d) = *N* intake-fecal *N*-urine *N*. Nitrogen utilization (%) = (*N* retention/*N* intake) × 100.

### 2.5. Analysis of Plasma Metabolites and AAs

Plasma total protein (TP, Method: biuret), albumin (Alb, Method: bromocresol green), glucose (Glu, Method: glucose oxidase), urea (Method: two-point dynamic) and ammonia (Method: two-point dynamic) concentrations were measured using a fully automatic biochemical analyzer (HT82-BTS-330, Xihuayi Technology Co., Ltd., Beijing, China). Commercial acid kits from Nanjing Jiancheng Institute of Bioengineering (Nanjing, China) was used to analyze the plasma nitric oxide (NO, Method: nitrate reductase; Kit number: HY-60018), total antioxidant capacity (T-AOC, Method: 2, 2′-azino-bis (3-ethylbenzothiazoline-6-sulfonic acid); Kit number: HY-60021), catalase (CAT, Method: visible light; Kit number: HY-60015), total superoxide dismutase (T-SOD, Method: hydroxylamine; Kit number: HY-60001), malondialdehyde (MDA, Method: thiobarbituric; Kit number: HY-60003) and glutathione (GSH, Method: colorimetric; Kit number: HY-60006) concentrations. The concentration of plasma amino acids in plasma samples was measured using high-performance liquid chromatography-liquid chromatography-mass spectrometry (HPLC-LCMS/MS API3200 Q-TRAP, Thermo Fisher Scientific, Waltham, MA, USA).

### 2.6. Statistical Analysis

Before analyses, the normality of all data was tested using Minitab 17 (Minitab, Inc., State College, PA, USA, 2014) and all the data were in accordance with normal distribution. Thereafter, the data were analyzed using the mixed procedure of SAS 9.4 (SAS Institute, Cary, NC, USA) with repeated measures. The statistical model included initial variables (obtained at 0 weeks) as covariates and treatments, weeks and their interactions as fixed effects, as well as the random effects of blocks and bulls within blocks. Polynomial orthogonal contrasts were used to analyze the linear and quadratic effects of increasing NCG supplementation on each variable [24]. Significant differences were declared at *p* ≤ 0.05, and trends were defined at 0.05 < *p* ≤ 0.10.

## 3. Results

### 3.1. Growth Performance

A linear increase in ADG (*p* < 0.05) and FCR (*p* < 0.05) were observed with increasing doses of NCG (Table 2). Bulls fed NCG tended to show a quadratic increased ADG (0.05 < *p* < 0.10). Moreover, M bulls had a highest ADG among treatments. No differences (*p* > 0.10) in feed intake of DM, CP, NDF and ADF were observed among treatments.

### 3.2. Apparent Digestibility and N Metabolism

Table 3 shows the apparent digestibility and N metabolism results. No difference (*p* > 0.10) was observed in the apparent digestibility of DM, NDF, and ADF. However, the CP digestibility (0.05 < *p* < 0.10) tended to show a linear increase with increasing doses of NCG. In addition, bulls fed NCG tended to show a linear decrease in the fecal N contents (0.05 < *p* < 0.10) and urinary N (0.05 < *p* < 0.10). A linear increase in N retention (*p* < 0.05) and N utilization (*p* < 0.05), a quadratic increase in N retention (*p* < 0.05) and N utilization (*p* < 0.05) were observed with increasing doses of NCG. Moreover, M bulls had a highest N utilization among treatments.

### 3.3. Plasma Metabolite

As presented in Table 4, a linear decrease and a quadratic decrease in plasma urea concentrations (*p* < 0.05) and ammonia (*p* < 0.05) were observed with increasing doses of NCG. Conversely, a quadratic increase in plasma NO concentrations (*p* < 0.05) was observed with increasing doses of NCG. Additionally, M bulls had highest plasma urea and ammonia concentrations among treatments. Regarding plasma antioxidant variables, T-AOC, CAT, T-SOD, MDA, and GSH concentrations were not different (*p* > 0.10) among treatments.

### 3.4. Plasma Amino Acids

As shown in Table 5, regarding the essential amino acid (EAA) results, plasma Arg concentrations (*p* < 0.05) and Ile (*p* < 0.05) linearly and quadratically increased with increasing doses of NCG. Moreover, bulls fed NCG showed a linear increased Val (*p* < 0.05) and total essential amino acid (TEAA) (*p* < 0.05). Regarding the nonessential amino acid (NEAA) results, plasma Ala (*p* < 0.05), Glu (*p* < 0.05), Ser (*p* < 0.05) and total nonessential amino acid (TNEAA) (*p* < 0.05) concentrations linearly and quadratically increased with increasing doses of NCG. Nevertheless, no difference (*p* > 0.10) was observed in other amino acids among treatments.

## 4. Discussion

This study is the first to investigate the effect of dietary supplementation with NCG on growth performance, apparent digestibility, nitrogen metabolism and plasma metabolites in Holstein bulls. In the present study, no difference in DM, CP, NDF, and ADF intake was noted among treatments, indicating that NCG did not affect the bulls’ willingness to eat. This result is in agreement with the findings of Gu [11] and Chacher [12], who found that feeding NCG to cows did not affect their DMI. Furthermore, we also observed that ADG and feed efficiency in the M group were higher than those in the other groups. This phenomenon may be due to the addition of NCG reducing the excretion of fecal and urine nitrogen in bulls, thus increasing the total digestibility of protein, and ultimately promoting protein deposition in muscles [25]. Furthermore, the concentrations of plasma Arg and Glu in M were higher than in H, indicating that feeding 40 mg/kg NCG could better promote the absorption of amino acids in bulls, and ultimately lead to higher protein synthesis. These results indicate that NCG can increase the growth performance of bulls. In line with our results, previously published literature has reported that lambs fed NCG showed improved growth performance compared with lambs not fed NCG [10]. Yang et al. (2013) demonstrated that weaning piglets fed NCG showed enhanced growth performance via regulation of the expression of intestinal nutrient transporters, thereby promoting the absorption of nutrients [26].

In the current study, CP digestibility increased in NCG-fed bulls, indicating that NCG may have the potential to improve CP digestion in the rumen and small intestine of bulls, thereby causing increased nitrogen utilization in NCG fed bulls. To the best of our knowledge, NCG fed bulls increased endogenous arginine production through efficient regulation of urea cycle and converted ammonia in blood into urea, thus reducing the content of fecal nitrogen and urine nitrogen, which in turn resulted in a higher CP digestion [27]. Indeed, the decrease in the contents of fecal nitrogen and urine nitrogen in NCG-fed bulls in the present study demonstrated this result. The contents of fecal nitrogen and urine nitrogen are strongly correlated with digestion and absorption of amino acids by the small intestine [28]. Interestingly, Sun et al. (2017) found that dietary supplementation with NCG promoted endogenous arginine synthesis in the intestine of sheep, thus further promoting the digestion and absorption of other amino acids [29]. This might explain why the contents of fecal nitrogen and urine nitrogen decreased in the present study. This result is in agreement with previous studies, which demonstrate that nitrogen utilization of weaning piglets increased in NCG-fed pigs [26].

Blood ammonia is a metabolic waste product of various tissues and amino acids in the body, and its increase will affect the health of animals [30]. It is a positive finding that plasma ammonia concentration decreased after feeding NCG to bulls, indicating that addition of NCG is beneficial to the health of bulls. A possible cause for the decreased plasma ammonia concentration is that NCG has the ability to promote the conversion of plasma ammonia to amino acids [12]. Similar results reported that NCG can treat neonatal hyperammonemia [31]. Plasma urea concentration depends on plasma ammonia concentration. Indeed, in the present study also found that plasma urea concentration decreased. Plasma urea is a metabolic product coming from protein in ruminants [32], which can reflect the state of nitrogen metabolism and the balance of the amino acids and is known to be negatively correlated with nitrogen utilization in ruminants [33]. Therefore, the decrease in the plasma urea concentration may be due to the increased nitrogen utilization in NCG-fed bulls. Consistent with our results, previous research found that a decrease in blood urea concentration in cows fed increasing doses of NCG [12]. Furthermore, our results also showed that bulls fed NCG exhibit an increased plasma NO concentration. The reason for this increase may be that the addition of NCG could promote the synthesis of endogenous arginine, ultimately leading to a higher plasma NO concentration [34].

The contents of free amino acids in blood can reflect the balance and metabolism of amino acids in animals. The free blood amino acid content will be correspondingly reduced when the amino acid content in the diet is insufficient [5]. Regarding the results of plasma free amino acids in the present study, although plasma Arg concentration in NCG-fed bulls was increased compared with that of CON bulls, plasma Arg concentration in M bulls was higher than in H bulls (M vs. H, 205 µmol/L vs. 174 µmol/L). This might be attributed to the plasma Glu concentration in the M group which was lower than in the H group (M vs. H, 149 µmol/L vs. 130 µmol/L). Previous studies have demonstrated that the increase in plasma Glu concentration promotes the synthesis of Arg [35]. In line with our findings, Gu et al. [11] found that cows fed 20 g of NCG/d showed a higher plasma Arg concentration than cows fed 40 g of NCG/d. In the present study, bulls fed NCG showed increased plasma Ile and Val concentrations compared with CON bulls, which indicates that the addition of NCG might promote protein synthesis in bulls. Furthermore, we observed that plasma TEAA, Ala, Glu, Ser, Cys, Asp and TNEAA concentrations increased in NCG-fed bulls. These increased concentrations of free amino acids can activate the mammalian rapamycin target protein complex l pathway, which provides an essential material source for animal protein deposition and is conducive to amino acid balance and protein deposition [36]. Therefore, the present results indicated that the addition of NCG is helpful for protein synthesis in the body, thereby improving growth of bulls. However, the mechanism of action by which NCG changes free amino acid levels in blood still needs further study.

## 5. Conclusions

As hypothesized, the addition of NCG resulted in increased plasma Arg, Ile, Val, TEAA and TNEAA concentrations, which in turn resulted in a higher *N* utilization and CP digestibility for growth and, therefore, higher ADG in NCG-fed bulls. Furthermore, the 40 mg/kg supplemental NCG treatment was optimal to improve growth of fattening Holstein bulls.

## Figures and Tables

**Table 1 animals-11-00126-t001:** Ingredients and nutrient compositions of the dietary treatments.

Item	Amount
Ingredient composition, % of DM	-
Corn grain	32.4
Peanut hull	15.0
Soybean hull	10.0
Distillers Dried Grains with Solubles	8.0
Corn gluten feed	15.0
Corn germ meal	12.0
Molasses	4.5
Salt	0.8
Limestone	1.0
Magnesium oxide	0.5
Sodium bicarbonate	0.6
Mineral-vitamin premix ^1^	0.2
Chemical composition, % of DM	-
OM	90.5
CP	11.3
NDF	31.9
ADF	17.0
EE	3.3
Ca	0.8
P	0.4

^1^ The premix provided the following per kilogram of the diet: VA 2500 IU, VD 500 IU, VE 10 IU, Fe 10 mg, Cu 15.0 mg, Mn 20 mg, Zn 25 mg, I 0.50 mg, Co 0.10 mg. The percentage of minerals in the premix was 28.3%; the percentage of vitamins in the premix was 2.60%.

**Table 2 animals-11-00126-t002:** Effect of supplementation with N-carbamylglutamate (NCG) on the growth and intake of fattening Holstein bulls.

Item ^1^	Treatment ^2^				SEM	*p*-Value	
	CON	L	M	H		Linear	Quadratic
Growth performance							
ADG, kg/d	1.03	1.19	1.40	1.26	0.081	0.021	0.072
FCR	11.92	9.22	7.76	8.62	0.832	0.006	0.045
Intake kg/d							
DM	10.9	10.7	10.6	10.8	0.099	0.324	0.107
CP	1.23	1.22	1.20	1.22	0.016	0.590	0.537
NDF	3.48	3.44	3.38	3.45	0.050	0.537	0.302
ADF	1.84	1.82	1.79	1.83	0.027	0.549	0.314

^1^ ADG, average daily gain; FCR, feed conversion ratio (kg of DMI/kg of ADG); DM, dry matter; CP, crude protein; NDF, neutral detergent fiber; ADF, acid detergent fiber. ^2^ CON, control diet; L, supplementation with 20 mg/kg BW NCG; M, supplementation with 40 mg/kg BW NCG; H, supplementation with 80 mg/kg BW NCG.

**Table 3 animals-11-00126-t003:** Effect of supplementation with N-carbamylglutamate (NCG) on apparent digestibility and N metabolism in fattening Holstein bulls.

Item ^1^	Treatment ^2^				SEM	*p*-Value	
	CON	L	M	H		Linear	Quadratic
Digestibility %							
DM	68.2	66.8	70.2	68.0	0.999	0.570	0.673
CP	64.3	63.8	67.7	65.8	1.066	0.096	0.518
NDF	36.9	35.0	39.9	35.4	2.093	0.991	0.534
ADF	26.8	24.4	28.8	25.3	2.061	0.989	0.784
N metabolism (g/d)							
N intake	197	195	192	196	1.796	0.324	0.107
Fecal N	70.5	70.6	62.0	67.0	2.265	0.074	0.302
Urinary N	80.2	77.0	68.3	77.6	1.816	0.058	0.003
N retention	46.6	47.5	61.5	51.0	2.227	0.014	0.022
N utilization %	23.6	24.3	32.1	26.1	1.135	0.008	0.007

^1^ DM, dry matter; CP, crude protein; NDF, neutral detergent fiber; ADF, acid detergent fiber; *N*, nitrogen. ^2^ CON, control diet; L, supplementation with 20 mg/kg BW NCG; M, supplementation with 40 mg/kg BW NCG; H, supplementation with 80 mg/kg BW NCG.

**Table 4 animals-11-00126-t004:** Effect of supplementation with N-carbamylglutamate (NCG) on the plasma biochemistry and antioxidants in fattening Holstein bulls.

Item ^1^	Treatment ^2^				SEM	*p*-Value	
	CON	L	M	H		Linear	Quadratic
Plasma biochemistry							
TP g/L	67.9	68.3	70.5	69.8	1.174	0.145	0.630
Alb g/L	29.0	28.5	28.2	29.1	0.742	0.976	0.339
Glu mmol/L	5.08	5.16	5.07	5.25	0.154	0.551	0.766
Urea mmol/L	2.54	2.21	1.99	2.21	0.110	0.022	0.020
Plasma ammonia µmol/L	64.8	47.0	34.4	46.0	0.830	<0.001	<0.001
Plasma antioxidant							
NO µmol/L	38.5	39.0	50.6	38.8	2.468	0.265	0.021
T-AOC U/ml	8.98	9.01	8.90	9.03	0.712	0.992	0.939
CAT U/ml	69.1	66.3	66.6	65.1	2.692	0.344	0.816
T-SOD U/ml	74.4	76.7	78.9	73.8	3.499	0.985	0.295
MDA nmol/ml	3.92	3.79	3.86	3.87	0.098	0.879	0.499
GSH µmol/L	8.07	7.43	7.65	7.66	0.218	0.312	0.153

^1^ TP, total protein; Alb, albumin; Glu, glucose; NO, nitric oxide; T-AOC, total antioxidant capacity; CAT, catalase; T-SOD, total superoxide dismutase; MDA, malondialdehyde; GSH, glutathione. ^2^ CON, control diet; L, supplementation with 20 mg/kg BW NCG; M, supplementation with 40 mg/kg BW NCG; H, supplementation with 80 mg/kg BW NCG.

**Table 5 animals-11-00126-t005:** Effect of supplementation with N-carbamylglutamate (NCG) on plasma amino acids in fattening Holstein bulls.

Item ^1^	Treatment ^2^				SEM	*p*-Value	
	CON	L	M	H		Linear	Quadratic
EAA µmol/L							
Arg	142	155	205	174	6.663	<0.001	0.004
His	85.6	86.1	90.9	86.9	1.676	0.250	0.196
Leu	136	136	135	140	2.114	0.265	0.247
Ile	78.3	76.5	101	86.7	1.303	<0.001	<0.001
Lys	128	129	133	132	2.020	0.110	0.863
Met	33.5	34.3	35.7	32.0	1.079	0.517	0.047
Phe	38.2	40.3	41.0	41.3	1.001	0.036	0.377
Thr	76.8	75.6	73.9	75.7	1.157	0.352	0.221
Val	196	189	206	209	3.777	0.003	0.152
TEAA	935	911	1007	983	12.95	<0.001	0.977
NEAA µmol/L							
Ala	182	183	212	197	2.643	<0.001	0.008
Glu	109	125	149	130	1.660	<0.001	<0.001
Pro	78.3	79.2	79.0	78.2	1.743	0.946	0.655
Gly	407	391	394	402	5.510	0.614	0.045
Ser	89.6	104	111	106	2.497	<0.001	0.001
Tyr	57.2	58.1	59.6	58.2	1.259	0.433	0.375
Cys	0.64	0.71	0.72	0.73	0.231	0.057	0.368
Asp	16.9	18.1	19.4	18.5	0.616	0.045	0.112
TNEAA	946	971	1044	1017	7.833	<0.001	0.003

^1^ EAA, essential amino acid; TEAA, total essential amino acid; NEAA, nonessential amino acid; TNEAA, total nonessential amino acid; Arg, arginine; His, Histidine; Leu, Leucine; Ile, Isoleucine; Lys, Lysine; Met, Methionine; Phe, Phenylalanine; Thr, Threonine; Val, Valine; Ala, Alanine; Glu, Glutamic acid; Pro, Proline; Gly, Glycine; Ser, Serine; Tyr, Threonine; Cys, Cysteine; Asp, Aspartic. ^2^ CON, control diet; L, supplementation with 20 mg/kg BW NCG; M, supplementation with 40 mg/kg BW NCG; H, supplementation with 80 mg/kg BW NCG.

## Data Availability

No new data were created or analyzed in this study. Data sharing is not applicable to this article.

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
