# Peer review of "Effects of Dietary N-Carbamylglutamate on Growth Performance, Apparent Digestibility, Nitrogen Metabolism and Plasma Metabolites of Fattening Holstein Bulls"

_animals, 2021, doi:10.3390/ani11010126_

Round 1

Reviewer 1 Report

Manuskrypt jest interesujÄ…cy i ma potencjaÅ‚ do zastosowania, ale autorzy popeÅ‚nili kilka błędów.

Jak to się stało? na przykład w linii 30 (P = 0,013) i linii 34 (P <0,001). Tabela 3. P <0,05. Bez konsekwencji (konsekwencja).

PiÅ›miennictwo: nieprawidÅ‚owe cytowanie, brak numerów DOI.

Author Response

Response to Reviewer 1 Comments

animals-1032031

Effects of dietary N-carbamylglutamate on growth performance, apparent digestibility, nitrogen metabolism and plasma metabolites of fattening Holstein bulls

Dear Reviewer:

We are truly grateful to your critical comments and thoughtful suggestions for our manuscript. They are really helpful and based on these comments and suggestions, we have revised the manuscript carefully. Revised portions have been marked in yellow in the revised manuscript. In the following pages are our point-by-point responses to your comments/suggestions. Please feel free to contact us if there is any question and we are very willing to improve our manuscript at all times until you are satisfied.

Best regards,

Yonggen Zhang

Address: College of Animal Sciences and Technology, Northeast Agricultural University, Harbin, 150030, P. R.

Tel: +86 0451 5519 0840

Fax: +86 0451 5519 0840

E-mail: zhangyonggen@sina.com

Reviewer 2 Report

This paper aimed to investigate the effect of of adding different levels of N-carbamylglutamate on growth performance, nutrient digestibility, nitrogen metabolism, plasma metabolites and blood amino acids of fattening Holstein bulls. Overall the level of English is sufficient and most sentences read well. The main points for improvement are:

-Include In abstract values for average daily gain, feed efficiency and CP digestibility.

-Describe percentages of mineral/vitamins in the premix at the bottom of Table 1.

-Define ADG, FCR, DM, CP, NDF, ADF in Table 2.

-Define DM, CP, NDF, ADF, N in Table 3

-Define variables in column 1 in Table 4.

-Define variables indicated in the first column of Table 5.

-Indicate the average daily gain of bulls considered to formulate the diets.

-In statistical analyses indicate if all data presented normal distribution

-…   The unavailable amino acids were converted into urea, which was discharged into the environment through fecal nitrogen and urine nitrogen, by the ornithine cycle (REFERENCE)

-Authors believe that the addition of NCG benefits the health of bulls. What indications they have, besides plasma ammonia concentration reduction, to affirm this.

-A deeper explanation of the decreased average weight gain in H bulls compared to M bulls would be useful.

-Feed per kg of weight gain is 1.5 kg greater in L as compared to M group, yet no statistical differences were found. Could you recheck these analyses.

Other observations are depicted in the text.

Author Response

Response to Reviewer 2 Comments

animals-1032031

Effects of dietary N-carbamylglutamate on growth performance, apparent digestibility, nitrogen metabolism and plasma metabolites of fattening Holstein bulls

Dear Reviewer:

We are truly grateful to your critical comments and thoughtful suggestions for our manuscript. They are really helpful and based on these comments and suggestions, we have revised the manuscript carefully. Revised portions have been marked in yellow in the revised manuscript. In the following pages are our point-by-point responses to your comments/suggestions. Please feel free to contact us if there is any question and we are very willing to improve our manuscript at all times until you are satisfied.

Best regards,

Yonggen Zhang

Address: College of Animal Sciences and Technology, Northeast Agricultural University, Harbin, 150030, P. R.

Tel: +86 0451 5519 0840

Fax: +86 0451 5519 0840

E-mail: zhangyonggen@sina.com

Reviewer 3 Report

The aim of the study is the evaluation of the effect of supplementation N-carbamylglutamate for bulls for fattening to improve nitrogen utilisation and growth of animals.

The article is quite well prepared and I have only some question and recommendations.

Material and method:

Please specified better the duration of the experiment. You wrote that there were two weeks of adaptation period, but it is not clear how long was the duration of supplementation. BW was evaluated 0, 4 and 7 weeks of the trial period – including adaptation period or not?

Lines 139-140 – „A commercial thiobarbituric acid kit from Nanjing Jiancheng Institute of Bioengineering was used to analyze the plasma nitric oxid, total antioxidant capacity, catalase, total soperoxiddismutase, malondialdehyde and GSH.“ This is probably not true that you use one test for so many parameters. Please correct it. Which tests were used for other biochemical parameters (TP, Alb, Glu, Urea, ammonia)?

Results

You show in the text many times the level of statistical significance; nevertheless in different forms as P=... or P<..... Because there are all P-values in the tables it is redundant to show all the data again in the text. It sufficient to show only P<..... and not in all cases (e.g. significant dif......)

Table 3: It is quite surprising that you found statistically significant linear effect of the dose to fecal N, Urinary N, N retention and N utilization. In all these parameters were the highest or the lowest values for M-groups, which is the middle dose. Is it correct? Could you explain these linear relations?

Discussion

Line 232-234: “The unavailable amino acids were synthesized into urea, which was discharged into the environment through fecal nitrogen and urine nitrogen, by ornithine cycle.”  This sentence is completely unclear. What do you want to say? Amino acids are not synthesized into urea (urea is synthesized from ammonia, CO2 , Asp in ornithin cycle). Please check and correct this sentence.

The discussion concerning the tested levels is missing. The best results were obtained with middle dose (40 mg/kg BW). Do the authors have any information concerning the safety of the product? What about the doses tested in dairy cattle, which were optimal? What about the doses tested in other animal species?

The conclusion: „Overall, the present study provides a data reference for the widespread of NCG in beef cattle production,“ without more deep analysis of tested doses is not correct. Significant improvement was found mainly in middle dose. There could quite narrow limit for safety.

Author Response

Response to Reviewer 3 Comments

animals-1032031

Effects of dietary N-carbamylglutamate on growth performance, apparent digestibility, nitrogen metabolism and plasma metabolites of fattening Holstein bulls

Dear Reviewer:

We are truly grateful to your critical comments and thoughtful suggestions for our manuscript. They are really helpful and based on these comments and suggestions, we have revised the manuscript carefully. Revised portions have been marked in yellow in the revised manuscript. In the following pages are our point-by-point responses to your comments/suggestions. Please feel free to contact us if there is any question and we are very willing to improve our manuscript at all times until you are satisfied.

Best regards,

Yonggen Zhang

Address: College of Animal Sciences and Technology, Northeast Agricultural University, Harbin, 150030, P. R.

Tel: +86 0451 5519 0840

Fax: +86 0451 5519 0840

E-mail: zhangyonggen@sina.com

Reviewer 4 Report

Dear Editor,

I have reviewed the manuscript Animals 1032031 entitled “effects of dietary n-carbamylglutamate on growth performance, apparent digestibility, nitrogen metabolism and plasma metabolites of fattening Holstein bulls.

The study is well written and correspond to the objectives research field of the journal. The objectives of the study are clear, and the results are well presented. I have some concerns regarding supplementation, sampling, statistical analysis and validity of some data presented. These concerns are:

  1. At the highest does (80 mg/kg BW/d) bulls were given 32 g/d of NCG. This is a large supplementation of an AA analog. At the same time, bulls on the control were not supplemented with an equivalent. Providing a Placebo (ex glutamic acid, or arginine) may have elucidated the effect of NCG supplement from other possible AA deficiency.
  2. Methodology of N excretion calculation is not well described and point incorrect sampling. Mentioned in L105-106 and 113-114, spot sampling of fecal and urinary excretions were collected are 0600 and 1800 on weeks 0, 4 and 7 and composited by bull used to calculate N excretion. This has two potential problems:
    1. Compositing samples of week 0 (before treatment) with weeks 4 and 7 (after treatment) is wrong and does not isolate the effect of treatment.
    2. With spot sampling you do not have total volume excreted and therefore you cannot calculate N excretion (g/d) through feces or urine.
  3. Two statistical analysis were conducted on the same data set. The orthogonal contrast cannot be coupled with multiple comparison of means (Tukey’s test) on the same data. This lead to confusion, and misleading in data interpretation. It is highly recommended to choose one type of analysis and use it on all data in the manuscript. This requires major revision of the written text.
  4. There is incoherence is some data presented. For example, CP digestibility value should be similar to 100- (% fecal-N/N intake) which is not the case at all. Same for N intake and CP intake.
  5. Number of animals per treatment (6) is quite low. Reference to similar studies that used similar or lower number of animals in the same experimental setting (randomised design) should be cited

Specific comments

L25 blocked

L33 spell out NO

L38 remove CP digestibility as plasma level of AA does not affect digestibility

L38 growth and ADG imply the same thing, use one of them in the abstract

L51 remove semiessential proteinogenic

L52-54. Please reword this phrase, it does not provide accurate information in the current form

L56-57 This phrase can be removed. It is irrelevant to the text

L61 Please explain what does it mean “perform functions through the rumen”

L63 lactating lambs?

L61 Please explain what does it mean “improved milk quality and lactation performance”

L65-66 Please explain what does it mean “improved protein metabolism efficiency and balanced amino acids”

L71 Please remove « in feed »

L82 remove « similar »

L88 add details on the concentrates used as carrier of NCG

L103 what is frequency of sampling?

L122 Please add the AOAC procedure number for each type of analysis

L125 Please indicate how did you measure fecal and urinary output?

L227 Provide mechanism rxplaining how NCG increases CP digestibility

L232 please explain « unavailable AA »

L234-236 Please explain the relationship between endogenous arginin and fecal and urinary N decrease in bulls consuming NCG

L239-241 GU et al. 2018 did not say that, please check your referencing

L246 please provide references for this statement

L242-252 Plasma urea and plasma NH3 should be discussed together rather than independently, as concentration of urea depends on NH3. Please correct accordingly.

L248 The statement ”plasma urea is a metabolic outcome of protein and amino acids” is very general, please be more specific

L254-262 The effect of NCG on NO was observed only in the M treatment. Therefore, the discussion in this section should be speculative, as concentreation of NO in the L and H treatments, as well as T-AOC were not supportive.

L263-264 This is a very general statement that does not provide any information. Should be removed.

L265 A sentence should not start with an abbreviation. Please reword. Same in L277

L265-270 This should be in the introduction rather than discussion section, where you should discuss the findings of the study.

L277-279 this is irrelevant, please remove or condense.

L290-293 This was not discussed in the manuscript.

Author Response

Response to Reviewer 4 Comments

animals-1032031

Effects of dietary N-carbamylglutamate on growth performance, apparent digestibility, nitrogen metabolism and plasma metabolites of fattening Holstein bulls

Dear Reviewer:

We are truly grateful to your critical comments and thoughtful suggestions for our manuscript. They are really helpful and based on these comments and suggestions, we have revised the manuscript carefully. Revised portions have been marked in yellow in the revised manuscript. In the following pages are our point-by-point responses to your comments/suggestions. Please feel free to contact us if there is any question and we are very willing to improve our manuscript at all times until you are satisfied.

Best regards,

Yonggen Zhang

Address: College of Animal Sciences and Technology, Northeast Agricultural University, Harbin, 150030, P. R.

Tel: +86 0451 5519 0840

Fax: +86 0451 5519 0840

E-mail: zhangyonggen@sina.com

Reviewer 5 Report

A good piece of research work which reports a new finding in the quest to improve Nitrogen utilization by bulls through feeding of N-carbamylglutamate. I have made a few minor edits in the text (see attached) to help the authors improve on their communication and the quality of the manuscript.

Author Response

Response to Reviewer 5 Comments

animals-1032031

Effects of dietary N-carbamylglutamate on growth performance, apparent digestibility, nitrogen metabolism and plasma metabolites of fattening Holstein bulls

Dear Reviewer:

We are truly grateful to your critical comments and thoughtful suggestions for our manuscript. They are really helpful and based on these comments and suggestions, we have revised the manuscript carefully. Revised portions have been marked in yellow in the revised manuscript. In the following pages are our point-by-point responses to your comments/suggestions. Please feel free to contact us if there is any question and we are very willing to improve our manuscript at all times until you are satisfied.

Best regards,

Yonggen Zhang

Address: College of Animal Sciences and Technology, Northeast Agricultural University, Harbin, 150030, P. R.

Tel: +86 0451 5519 0840

Fax: +86 0451 5519 0840

E-mail: zhangyonggen@sina.com
